# Positive Psychology and Philosophy-as-Usual: An Unhappy Match?

**Josef Mattes**

Faculty of Mathematics, University Wien, 1010 Wien, Austria; a08401972@unet.univie.ac.at

**Abstract:** The present article critiques standard attempts to make philosophy appear relevant to the scientific study of well-being, drawing examples in particular from works that argue for fundamental differences between different forms of wellbeing (by Besser-Jones, Kristjánsson, and Kraut, for example), and claims concerning the supposedly inherent normativity of wellbeing research (e.g., Prinzing, Alexandrova, and Nussbaum). Specifically, it is argued that philosophers in at least some relevant cases fail to apply what is often claimed to be among their core competences: conceptual rigor—not only in dealing with the psychological construct of flow, but also in relation to apparently philosophical concepts such as normativity, objectivity, or *eudaimonia*. Furthermore, the uncritical use of so-called thought experiments in philosophy is shown to be inappropriate for the scientific study of wellbeing. As an alternative to such philosophy-as-usual, proper attention to other philosophical traditions is argued to be promising. In particular, the philosophy of ZhuangZi (a contemporary of Aristotle and one of the most important figures in Chinese intellectual history) appears to concord well with today's psychological knowledge, and to contain valuable ideas for the future development of positive psychology.

**Keywords:** well-being; flourishing; eudaimonia; method of cases; conceptual analysis; ZhuangZi; Aristotle; world philosophy; transcultural philosophy; dual-process models

## 1. Introduction

### 1.1. Background

The study of well-being is a currently a highly active area of research, with contributions coming from both philosophy and psychology.

In psychology in particular, about twenty years ago the "positive psychology" movement started off with the goal of redressing the bias towards the negative and pathological, which is widely found in psychology. Even though a similar goal had already been pursued for several decades by humanistic psychology, the need for a new positive psychology was felt because of the perceived shortcomings of humanistic psychology in terms of rigorous science.

> We well recognize that positive psychology is not a new idea. It has many distinguished ancestors, and we make no claim of originality. However, these ancestors somehow failed to attract a cumulative, empirical body of research to ground their ideas. [1] (p. 13)

This holds even more so with regard to the relationship of positive psychology with philosophy and religion. When, for example, Mihály Csíkszentmihályi tried to understand why some people managed to display serenity and keep their integrity and purpose despite the chaos of World War II, he observed that

> reading philosophy and dabbling in history and religion did not provide satisfying answers to that question. I found the ideas in these texts to be too subjective, to be dependent on faith or to be dubious assumptions; they lacked the clear-eyed



skepticism and the slow cumulative growth that I associated with science. [1] (p. 7)

Hence, being *scientific* is an essential part of the *raison d'être* of the field of positive psychology[1].

Recently, the focus of attention has shifted somewhat, so that the nature of the "positive" in positive psychology has become the primary subject of current debate (for example [3–8]), with contributions not only from psychologists but also from philosophers. In particular, Intelisano, Kraskol, and Luhmann [9] noted interest in integrating the two disciplines, added that philosophers have been interested in happiness and well-being since the Hellenic period, and asserted that "the terms 'happiness' and 'well-being' refer to highly distinct philosophical traditions but are used interchangeably by some researchers of both disciplines". [9] (p. 161f) These authors also referred to philosophical theories of well-being as being "normative" in contrast to philosophical theories of happiness, which they referred to as descriptive, with this distinction in turn apparently being different from that between hedonic and eudaimonic accounts in psychology [9] (p. 162). In this, Intelisano et al. evidently take it as a given that integrating philosophy and psychology is desirable or maybe even necessary, without stating *why* this should be the case. However, as Alexandrova [3] (p. xv) observed, "It is no good clamouring for a greater attention to philosophy if philosophy does not have much to offer". This sets the motivating questions for the present paper: What, if anything, has *philosophy* to offer to the *science* of well-being? What, if anything, does the answer to the previous question tell us about the practice of philosophy?

The answers to the previous questions depend of course to some extent on what philosophy is. A comprehensive discussion of this is beyond what is possible in this paper; is seems nevertheless safe to assume that conceptual clarification would be a major part of it, given that, on one hand, philosophers tend to see this as a skill central to their trade, and on the other hand this frequently constitutes the basis of philosophical criticisms of positive psychology: Statements such as 'Philosophers often prioritise conceptual rigour' [6] (p. 541), or "although conceptual analysis has come under pressure in recent years, it is still the standard procedure in mainstream analytic philosophy, to the extent that there is a standard procedure" [10] (p. 71) illustrate the first claim, while examples for the second are provided by [6] (p. 542) attesting "terminological disarray" in the field of wellbeing research, and [11] (p. S108) accusing it of being "riddled with conception confusion".

Many psychologists are indeed aware that terms such as "happiness", "wellbeing", and "*eudaimonia*" have been used in multiple ways (see for example Vittersø [8] (Tables 1 and 2 on p. 10f)) as well as of the difficulty in translating the ancient Greek term *eudaimonia* into contemporary English [8] (p. 7f). Thus, it would not be surprising if many positive psychologists take it for granted that philosophers contribute value through the necessary enforcement of conceptual rigour.

*1.2. The Present Work*

The purpose of the first part of the present paper is to challenge this presupposition. This is accomplished by pointing out important examples of failures by philosophers to apply conceptual rigour to positive psychology concepts like the contemporary term "flow", or even to philosophical terms like "normativity". The goal is not a complete review of the literature, but rather to use examples to raise reasonable doubt. In this, the emphasis will be on some discussions of putative differences between *eudaimonia* and "happiness". In addition, some of the current challenges to usual philosophical methodology are recalled, challenges which most psychologists may not be aware of. Finally, the paper very briefly sketches how an alternative to philosophy-as-usual[2] may be conceived in order to obtain a mutually beneficial relationship between philosophy and positive psychology.

The concern motivating this paper is the use and impact of philosophy in psychology in general, and especially in positive psychology. No less an analytic philosopher than Quine noted already in 1981 that "quality control is spotty in the burgeoning philosophical

press" [12] (p. 193); since then, things may not have improved much if at all. Admittedly, there are problems in the sciences as well, but they are at least widely recognized (compare Section 4 below), whereas for example the critical work of Machery on philosophical method (see below Section 3.2) is far less so—to the best of my knowledge it is not mentioned anywhere in the positive psychology literature.

Please note the question mark at the end of the title of this paper; there is no claim to have *proven* that positive psychology and philosophy-as-usual are incompatible. The aim is only to raise reasonable doubt as to the seeming rigour of philosophy-as-usual, for this providing examples as in the present paper should be sufficient. Please also note that there is no claim that the examples chosen are cases of particularly bad philosophy, they were rather chosen because they appeared in leading journals in another discipline. After all, even if, as the analytic philosopher Daniel Dennett [13] (ch. 4) asserted in arguing against criticisms of philosophy, "there is a lot of mediocre work done in every field", this does not imply that one should "concentrate on the best stuff". This would not be responsible, since "stuff" that is problematic (not-among-the-best) can still have a detrimental impact, in particular when it is published in prominent journals in another field (e.g., philosophical works appearing in psychology journals), where readers may be less able to critically appraise the problematic aspects.

A second guiding idea for the present work is that science also crucially relies on creativity, on generating ideas and on breaking out of old thinking habits; and that in this respect, philosophy can indeed be helpful—in particular if it is understood as *world philosophy* ("philosophy that goes beyond any single philosophical tradition in seeking to work out a satisfactory overall or partial philosophical view of things", as Slote [14] (p. 374) put it). Consequently, in an extended appendix[3] I draw on the ZhuangZi—and would, if space and time permitted, most likely also draw on Early Buddhism (this is work in progress, compare below endnotes 22 and 25)—because it seems to me to be more appropriate for positive psychology[4] compared with philosophy-as-usual. In this, the meta-philosophical attitude is *trans*cultural rather than *inter*cultural. This seems to differentiate the present work from many of the current cross-cultural research projects within positive psychology.

## 2. Conceptual Rigour through Philosophy-as-Usual?

### 2.1. Example 1: "Flow" and the Concordance Thesis

#### 2.1.1. Background

It is a widely (but far from unanimously) held belief that there is an important dichotomy between positive emotion and/or positive cognitive evaluation of one's life on the one hand, and "flourishing" on the other. Here, flourishing is often understood along the lines of Aristotle's concept of *eudaimonia* (which in turn is related to the ancient Greek *aretê*, which means *excellence* but is often (mis-)translated as "virtue"[5]. In addition, emotions and cognitions are often viewed as subjective forms of well-being, supposedly distinct from allegedly objective *flourishing*.

Recently, Kristjánsson [6] argued against one attempt by the philosopher Julia Annas to bridge this apparent divide, an attempt which was based on Aristotle's suggestion that virtuous activity is pleasant to the mature virtuous person. [17,18] According to Annas, this suggestion can be understood in terms of the psychological concept of flow:

> The virtuous person, like the expert in a practical skill, responds dynamically to challenges, but this is, we may think, experienced in action as a selfless kind of flow. [17] (p. 33)

Referring to work by Besser-Jones [19], Kristjánsson claimed that truly virtuous activities are not likely to produce flow, and in addition provided four examples that he believed to be counterexamples to what he called the "concordance thesis", the thesis that flourishing (Aristotelian *eudaimonia*, supposedly an objective version of wellbeing) and

happiness (positive affect and/or satisfaction with life, supposedly subjective forms of wellbeing), when correctly conceptualised, go hand in hand.

Indeed, as presented by Annas, Aristotelian *eudaimonia* is a sufficient condition of living happily, as it is conducive to experiencing flow in that

> virtuous activity, as opposed to merely self-controlled activity, is pleasant, not in involving extra feelings but in being unimpeded by contrary impulses, and in harmony with all of the person's thoughts and feelings. In the virtuous, virtuous activity can be thought of as an example of "flow" because it is an unforced expression of the person's reasoning and feelings, in harmony with the rest of her character and structured system of goals. [17] (p. 30)

Against this, Besser-Jones claimed that flow states demand a balance of challenge and skill, so that

> flow experiences occur when individuals engage in complex and challenging activities that test one's capacities [...] cognitive engagement is crucial, as part of the enjoyment lies in the exercise of her intellect—in the problem-solving, [19] (p. 96)

and concluded from this that most virtuous activities are not the kinds of things that generate flow experiences.

Along similar lines, and partly based on this, Kristjánsson asserted that

> most of the virtuous activities that Aristotle sees as flourishing–constituting are pretty dull and uninspiring in themselves [and] not likely to produce flow, [6] (p. 546)

which, according to Kristjánsson, provides a strong counterargument to the concordance thesis. In addition, Kristjánsson presented four supposed counterexamples to this concordance thesis: three of them purely fictional, the sole non-fictional one being Ludwig Wittgenstein. These examples will be discussed later, after exploring the conceptualizations of flow in the psychological literature, and their implications.

2.1.2. Disambiguating "Flow"

This section argues that the psychological literature uses the term "flow" to denote a number of different concepts. This observation is not entirely new, but seems to be widely ignored even though highly relevant to the present topics.

Mihály Csíkszentmihályi, the founder of what is today usually known as *flow theory*, a theory of intrinsic motivation which was based on the study of human activities that are not primarily driven by the expectation of external rewards, but were done rather for the sheer enjoyment of doing them. [20] Interview studies in various populations worldwide led him to the conclusion that the descriptions of what makes an experience enjoyable were similar even though "what the [interviewees] did to experience enjoyment varied enormously—the elderly Koreans liked to meditate, the teenage Japanese liked to swarm around in motorcycle gangs" [21] (p. 48). At first, he called this the *autotelic* experience, later "to use more accessible language" [22], *flow*. In [23], Csíkszentmihályi again identified flow with the autotelic experience and also stressed the importance of the *autotelic personality*, "those who have such flow experiences relatively often, *regardless of that they are doing*" [23] (p. 824, emphasis added). In fact, his studies "have suggested that happiness depends on whether a person is able to derive flow from whatever he or she does [24].

As noted in the introduction, early attempts in psychology to study human flourishing, for example in humanistic psychology, were criticised for lack of scientific rigour. Psychology as a science rightly places importance on empirical research and experimental validation of theories. In order to make the flow concept amenable to experiments it was necessary to operationalize the flow construct by constructing models of it. The simplest flow model describes flow as the state in which one is neither anxious nor bored because "skills" and "challenges" are perceived as being roughly equal, later models added more

states such as apathy (when skills and challenges match, but are both at a low level), or divide "anxiety" into anxiety proper, worry, and arousal as well as "boredom" into boredom, relaxation, and control. [25]

Kawabata and Mallett [26] pointed out a number of conceptual problems in the literature on flow, including problems with the notion of challenge as used in flow theory. Specifically concerning the skills–challenges balance, Landhäußer and Keller [27] noted that in most cases, researchers purportedly investigating flow actually investigated correlates and consequences of skills–demands compatibility—which is a precondition of flow, not the flow experience itself. They then pointed out that

> researchers seem to equalize the precondition of flow [...] with the experience itself [...] Because the association between the preconditions of flow and the experience itself is definitely not deterministic [...] this is problematic. [ . . . A] measure of skills–demands balance should not be used (or interpreted) as a measure of the flow experience per se.

Consistent with this, a meta-analysis [28] of the relationship between skills–demands balance and flow experiences revealed it to be only moderate. This fact is even more problematic since the skills–demands balance is often referred to as a characteristic of (rather than a precondition for) the flow experience, for example in [21]. In fact, even calling it a "precondition" seems inappropriate since it misleadingly suggests necessity for, rather than just raising the probability of, a flow experience happening; and also because flow conditions differ across models. [29] (p. 174) In short, flow is related to, but should not be defined via, perceived skills and demands during an activity.

In fact, the term flow has been used to refer to many related but distinct concepts, including:

1.  The experience of autotelic (intrinsically motivated) behaviour;
2.  Optimal experience, which in turn has several definitions including

    a.   Inner harmony ([30] (p. 24), [21] (p. 39)),
    b.   A 'complex and positive state characterized by deep involvement and absorption, supporting personal growth, well-being and optimal functioning in daily life' [31] (p. 3), and
    c.   The experience "where action becomes automatic and conscious thought seems to meld together with the action itself" [32] (p. 95);

3.  The experience of total involvement in what one is doing [33];
4.  Being "beyond boredom and anxiety" [20], or beyond arousal, control, relaxation, boredom, apathy, worry, and anxiety (e.g., [34]);
5.  Action following upon action according to an internal logic needing no recognizable external intervention by the actor, which is experienced as a unified flowing from moment to moment ([29] referring to [20] (p. 36));
6.  Skill–demand balance (e.g., [35]).

Rather than asking what is the "right" definition of flow, the issue here is which way of understanding "flow" is relevant to the discussion of wellbeing and *eudaimonia*. Here, it seems clear that Aristotle was interested in autotelic experience: *eudaimonia* is desired for its own sake (e.g., *Nicomachean Ethics* book 1.5), and the mature virtuous person enjoys inner harmony.

> Aristotle famously expresses this as the difference between the virtuous and the merely "encratic" or continent person, who acts in the same way as the virtuous, but is not yet virtuous, because acting virtuously comes up against his feelings and attachments. [18] (p. 67)

Therefore, the notion of flow relevant in the present context is that of autotelic optimal harmonious experience. Skill–demand balance, in particular, is only an enabling condition making flow more likely, but is not identical to the flow state [29] (p. 174).

2.1.3. Concept Clarification Failure

In light of the above discussion, Besser-Jones' and Kristjánsson's criticisms of Annas' work seem cases of the widespread confusion between enabling conditions of flow and the flow experience itself. There is no reason why a mature virtuous person should not be able to experience flow while "keeping one's promises, helping someone pick up papers she has dropped on the sidewalk, being a whistleblower, loaning money to a friend, raising money to help victims of natural disasters, and so on" [19] (p. 100). After all, as seen above, an autotelic personality "has to develop the ability to find enjoyment and purpose regardless of external circumstances" and "derive flow from whatever he or she does", hence also when picking up dropped papers. In fact, there is even a questionnaire that allows the measurement of flow proneness in maintenance activities (e.g., chores) by Ullén et al. [36]. Table 3 in that paper seems to suggest that in such activities flow proneness may be about as widespread as in work or leisure activities. This is not meant to deny that finding flow in activities which most of us most of the time find boring or anxiety-provoking may be difficult, but difficult does not imply impossible. All one can conclude is that autotelic personalities and mature virtuous persons are rare—but that is neither news nor a good reason to resign oneself to lower standards[6].

There are more problems with the accounts by Besser-Jones and Kristjánsson. In the former case, she gave the example of a rock climber and asserted that in all flow experiences "cognitive engagement is crucial, as part of the enjoyment lies in the exercise of [the] intellect". There may indeed be activities that involve heavy use of the intellect and still allow flow experiences (playing chess comes to mind as a possible example), but this seems already doubtful about rock climbing: do you really cognitively figure out which grip will bear your weight? Not to mention that many flow experiences, for example while dancing Tango Argentino, or while hiking a long-distance trail, seem to have nothing to do with exercising the intellect. In fact, flow has been discussed as prototypically non-deliberative "type 1" information processing (see below).

Overall, one has to conclude that the arguments provided by Besser-Jones and Kristjánsson against Annas' account of virtue and flow are unconvincing. This alone of course does not imply that Annas is right; a fuller evaluation would necessitate discussing the merits of the skills analogy. This, and topics such as the "unity of virtue" thesis, are nevertheless not the focus of the present paper. Nor is it intended as a full discussion of the flow concept(s): for more complete recent discussions[7] see for example [37] or [38]. What it does establish is that ambiguity in the flow concept could be found in the literature for decades already before Besser-Jones and Kristjánsson, two highly regarded philosophers, failed to analyse the concept of flow, despite it being central to their argument. This is relevant in the present context as this is not an exclusively inter-philosophical problem: Kristjánsson [6] appeared in a leading psychology journal.

Assuming that the above is right, it might conceivably be objected that "flow" is a psychological concept, so that it is unfair to expect philosophers to deal with it rigorously. Therefore, the following sections discuss examples of philosophers paying insufficient attention to the intricacies of what appear to be concepts central to the *philosophy* of wellbeing and which are claimed to be relevant to psychology as well.

*2.2. Example 2: "Objective Wellbeing"*

According to Kristjánsson [6] (p. 541), accounts of wellbeing typically "congeal into one of the two antitheses of *subjectivism* or *objectivism*"[8], with subjective accounts supposedly focusing either on pleasure (hedonic accounts) or on life satisfaction, whereas "objective ones tend to hark back to the Aristotelian notion of *eudaimonia*—constituting so-called *flourishing* or *eudaimonic* accounts". According to him, happiness and flourishing correspond reasonably well to subjective and objective wellbeing, respectively.

What is missing is a clear definition of "objective" and "subjective". Kristjánsson does seem to say that some components of models such as PERMA, PWB, or SDT (p. 542), including "relationships, engagements, the exercise of capabilities/virtues" (p. 543), are

objective at least if measured objectively, again without making explicit what is meant by measuring objectively. Nevertheless, the examples provided and the approving reference (p. 544) to the classification by MacLeod [39] may indicate that Kristjánsson has the same usage as MacLeod in mind. Unfortunately, that is quite problematic as it is based on a substantial assumption.

> Subjectivity can refer to somewhat differing things. A view can emphasise the importance for the well-being of positive feeling states such as happiness—someone has high well-being to the extent that they feel happy. Or, subjectivity can be characterised as the extent to which people define their own well-being in whatever way they choose, as opposed to having others define it by reference to external standards. These two aspects can be combined if it is assumed that when people are left to define their own well-being subjectively, experienced happiness is central to that definition. [39] (p. 1075)

This is a highly non-obvious assumption as it implies, for example, that people cannot base their subjective appraisal of their wellbeing on, say, post-traumatic growth (trauma is almost by definition not conducive to positive feeling states even after the fact), while many view such growth positively, at least in retrospect. Nor is it clear why an externally imposed standard would necessarily deserve being called objective: What if, for example, the standard is just one person's opinion? That is external, but surely not thereby any less subjective.

What is also confusing is that [6] (p. 542) approvingly referred to the "trenchant critique" by Keyes and Annas, of departures from Aristotle's original notion of eudaimonia, while himself ignoring their observation that the "distinction between objective and subjective does not map well onto Aristotle" [40] (p. 198)—which flatly contradicts the premise of his paper that there is such a distinction.

There is another problematic point about [6]: Its author professed to believe that counterexamples to his concordance thesis are "fairly common in daily life" (p. 549), but presented only fictious examples, except for one: Ludwig Wittgenstein. The relevant passage reads:

> It is almost de rigueur to invoke Wittgenstein as an example of an unhappy flourisher. His famous last words, 'Tell them that I've had a wonderful life' [...] are typically taken to mean that he considered himself to have flourished in life [...] However, by all accounts, he was a grumpy and miserable person with a serious happiness deficit.

I am surprised that this is presented as a counterexample to the concordance thesis, as such a counterexample would have to show a discrepancy between subjective happiness and objective flourishing. In the case of Wittgenstein, if he had a serious happiness deficit this might imply low subjective wellbeing (if he himself also perceived such a deficit), but I see nothing whatsoever relating to an objective form of wellbeing: After all, "Tell them that I've had a wonderful life" is in no way no less subjective than "Tell them that I've had a (highly) satisfactory life", and Kristjánsson classifies such life satisfaction statements as subjective wellbeing. Thus, if anything, his example speaks *against* his claimed dichotomy between flourishing and subjectivity.

### 2.3. Example 3: "Normativity"

It seems to be *en vogue* to lecture scientists about the allegedly unavoidable "value-ladenness" of the sciences in general, and supposedly intrinsically normative character of positive psychology in particular. This paper will not discuss the claim of general value-ladenness (see [41] for a critical discussion); it does argue that it is problematic that philosophical contributions tend to neglect providing the necessary clarification as to what concept of normativity they employ.

After all, there seem to be different ways in which the word can be understood. For example, laws of nature are in a sense normative: You will obey the law of gravity, whether

you want to or not, and whether you are aware of it or not. You have no choice, and human beings never had one. On the other hand, the belief that certain actions be better avoided because otherwise some god will put you in hell for eternity is normative in a different sense: at least in principle you can refuse to obey—unadvisable as this may seem to a believer—and it is only applicable since the time at which humans invented[9] moralizing gods.

Far less obvious is where the supposed normativity in various theories in contemporary moral philosophy might be coming from, and what exactly it might consist of. For example, what does the "ought" mean when [43] (p. 2) asserted that "Aristotle, Bentham, and Mill [ . . . were] interested in [ . . . ] what goals we ought to pursue". Is this "ought" a law of nature? A divine command? Kant's command? Prinzing's command? Is the Aristotle's "ought" the same as Bentham's? Is it an evaluation or a prescription? Is it not a pity that, for example, Alexandrova [3] writes a whole book full of claims of normativity in positive psychology, without the word "normativity" being found worthy of as much as a definition or an entry in the index? Similarly, Nussbaum [11] (p. A108)—approvingly quoted in the handbook of eudaimonic well-being [8] (p. 12)—accuses positive psychologists of "normative naïveté" without explaining anything about how she uses the term normative.

Fortunately, there are some positive examples. One is provided by Kristjánsson [6], who acknowledged in an endnote that there is an important difference between normativity understood as evaluativeness versus normativity as prescriptivity. This difference is crucial to positive psychology, as it is at least possible that the former understanding (evaluativeness) might be compatible with it being a *bona fide* science, whereas the latter clearly is not.

But do we not need to have ethical theories to tell us what to do? Here, the "need" is doubtful, and the "we" is even more so in view of the well-known false consensus effect [44,45]. It is helpful to recall that many normative theories seem highly problematic: Of particular relevance is of course Aristotle, as few today would agree with his views on women, "barbarians", and slaves. Moeller [46] also pointed out that not only do many assertions in the ethics literature seem bizarre, specifically discussing Bentham and Kant (e.g., quoting Kant as asserting that "A child that comes into the world apart from marriage is born outside the law (for the law is marriage) and therefore outside the protection of the law", p. 63) but noting that "Kant and Bentham have in common is their level of presumption. Both claimed to have identified *scientifically* the principles of good and evil".

How about today? Naturally, the current majority opinion looks reasonable to the current majority, but how about future views of current opinions? On the one hand one could argue that, if past ethical theories turned out to be untenable, one might expect the same of current ethical theories[10]. On the other hand, one might hope that contemporary philosophy is different as it supposedly emphasizes openness and tolerance—unfortunately, empirical evidence speaks against this supposition [47].

Another highly interesting area of philosophical and psychological research, unfortunately badly neglected in wellbeing research, concerns the nature of ethical expertise (if any) and practical application thereof. Thankfully, some philosophers and psychologists have performed research which appears relevant to this question. A number of studies showed not only that moral philosophers' judgements in ethical matters are influenced by irrelevant factors (e.g., [48,49]), but—maybe even more relevantly in the present context—they seem to steal at least as many books as other academics [50], and under some additional measures it does not seem like they act in accord with what their theories might prescribe either [51], results that have recently been replicated in a new study [52].

Thus, neither is it clear what is meant by claims that positive psychology is intrinsically normative, nor that what in practice has been the dominant strand of normative moral philosophies is fit for integration with science (see also Section 5.1). Overall, it appears that philosophers, by neglecting to point out all the intricacies and problems involved in claims of normativity, again fail at concept clarification.



*2.4. Example 4: "Eudaimonia"*

Those positing a dichotomy between forms of wellbeing often take recourse to the ancient Greek notion of *eudaimonia*, mostly in the form articulated in Aristotle's two writings on the subject: the *Nicomachean Ethics* (EN) and the *Eudemian Ethics* (EE). ([6] (p. 541), [9] (p. 188)). Would returning to the source [53] help reduce conceptual confusion in wellbeing research?

On first sight one might expect the answer to be "no", since Aristotle's ethical writings have spawned an immense literature containing serious disagreements on a number of interpretive issues. Aristotle has also been accused of being unclear and/or confused.[11] One pertinent issue, where Aristotle is often taken to be unclear, is the passage in book 10 of the EN, where he asserts the superiority of the contemplative life over the political life, with only the contemplative life realizing *teleia* (complete) *eudaimonia*; supposedly this is at odds with the rest of the EN as well as his attitude in his book *Politics*. There is of course the question of what level of precision and clarity can be expected in the first place. After all, Aristotle himself noted that the subject matter does not allow for the precision that one might hope for. Nor is the purpose of the EN and EE clear, since we do not know for which audience it was written (though it seems a fair guess that it was not for professional analytic philosophers trying to over-interpret minute details in a pair of unpolished lecture notes, compare [55,56]). In addition to all that, the works of Aristotle available to us today are only a small part of those listed in ancient lists of his works, and those that survived seem to have been stored in the basement of a house for well over a century—so one may wonder what importance was assigned to them by Aristotle's successors. Unfortunately, such complications often tend to get swept under the rug when philosophers write in the context of wellbeing research.

Does it matter for wellbeing research? Yes, if the nature of *eudaimonia* matters, and if Aristotle's explicit statements carry any weight. For example, he was explicit that *eudaimonia* is a goal in itself. So, when Thorsteinsen and Vittersø [7] (p. 520) refer to Aristotle's notion of a *telos* to motivate separating feelings into supposedly hedonic and supposedly eudaimonic ones[12], and at the same time want to distinguish them by the latter being involved in resource building, this fits badly: resources are by definition instrumental, thus for Aristotle they constitute at best an inferior *telos* rather than something eudaimonic.

Related to this seems to be the idea to develop one's potentials (ibid), or that the ideal behind *eudaimonia* (i.e., "*flourishing* in the fullest sense of the term") is "well-being as the realization of one's potential" [57] (p. 27). This seems odd: Surely, Genghis Khan realized an incredible amount of his potential, much more than pretty much everyone else. So Genghis Khan came close to the ideal eudaimonic life? I doubt that this is an intended consequence. Maybe Haybron and others mean only certain kinds of potential, but that raises the question of which ones ("positive potential" will not do, since we are trying to figure out what "positive" means). There is another problem: Realizing your potential is presumably easier if you have little potential—again, I doubt that this is intended[13].

## 3. Philosophy outside Its Proper Bounds

The previous section provided examples that shed doubt on the belief that psychologists can automatically assume philosophers' arguments will contribute to the scientific rigour that is at the heart of positive psychology. Now I put this into a larger context regarding philosophical methodology.

*3.1. Kraut's Oyster versus Nagel's Bat*

Kraut [58] argued for a well-being theory which "holds that: (A) well-being is composed of many goods; (B) all of them are experiential; but (C) pleasure is only one element of good experience". (p. 4) In this context, he is worried by an example from McTaggart, who

> asks us to compare two lives: the first he calls "oyster-like" because it has "very little consciousness and a very little excess of pleasure over pain"; the second is

that of a human being. His striking thesis is that a sufficiently longer oyster-like life is better than any shorter human life, no matter how wonderful the goods in the human life are. (p. 2)

Kraut finds this threatening, as in combination with (C) it comes close to implying that "attached as we are to the full range of human experiences, we should trade them away for a simple pleasure that is sufficiently long-lasting" (p. 5), a conclusion that Kraut finds "astounding" (p. 4) and which he obviously dislikes.

At the start of his discussion, Kraut states that

I will refer to the other creature in this comparison as "McTaggart's oyster" (not just oyster-like). To simplify matters further, I will assume that this oyster feels no pain and only the mildest of pleasures as it takes in nourishment. (p. 2)

His goal is to show that "certain human experiences, when they are of *sufficient* length (not any length, however small), are greater in prudential value than are the combined lifetime pleasures felt by McTaggart's oyster, however long it lives" (p. 21) Why would that be true? Here is what Kraut has to say:

We can have some notion of [the oyster's] inner life: it takes in nourishment and (we are assuming) it seeks more of the same because it has a pleasant sensation when it eats. We know through introspection what it is like to get pleasure from the taste of something. We can imagine what it is like to have nothing but that as one's form of consciousness, and can compare that kind of life to the much larger form of consciousness we are lucky enough to have, with respect to how good they are. [58] (p. 23)

What I find astounding is the confidence Kraut appears to place in his argument. He seems to take it for granted that he can imagine how it is to have nothing but an oyster's consciousness. That is a surprising claim, given that a few pages earlier he quoted approvingly Nagel's classical "bats" essay, which argued that "forms of experience quite alien to ours must therefore elude our grasp" [58] (p. 7). Given that bats are at least mammals, whereas oysters are not even vertebrates, what made Kraut believe that an oyster's consciousness is within his grasp whereas that of a bat is not? Even if one assumes that the oyster's experience is restricted to sensations of whatever "taste" is for a being with the sensory- and neuro-anatomy of an oyster, why should such "taste" be any easier for us to understand than the sensations the bat obtains from its ultrasound signals?[14].

However, even assuming that humans could introspect the experience of an oyster, there is a second major problem: humans are notoriously bad at dealing intuitively with very large (or very small) numbers. Presumably, there were few occasions for evolution to select in favour of those who were able to accurately deal with *very* large numbers like, say, the number of grains of sand in the Sahara desert[15]. This is relevant as Kraut claims one should never trade away what he believes to be quintessentially human experiences even for *sufficiently long-lasting* simple pleasures, i.e., no matter how long the pleasures last or how frequent they are: "one can be confident, through introspection, that the powerful emotional, visual, auditory, and cognitive experience one is having" (for example when sitting through a Wagner opera "with appreciation and understanding") "brings with it more prudential value than is available to any oyster, *however long it lives*" [58] (p. 212, emphasis added). Can one really introspect how it is to have certain experiences for longer than a human life (say, 200 years, a length which amounts to that of about 300,000 Wagner operas)? How it is to have them for much longer than a human life (say, 200,000 years)? Or for a Vast [16] number of years like, for example, 2,000,000,000,020,000,000,000, which is still a trifle compared to *however* long?

Supposedly, "experience assures us[17] [ . . . ] some amount (a sufficient amount) of one kind of good is preferable to *any* amount of other goods". (p. 170, emphasis added) What *experience* does Kraut have with goods that last for time stretches that vastly exceed human

life spans?[18] The belief that experiences and/or intuitions can settle this matter seems to me to be highly problematic.

*3.2. Relevance to Positive Psychology*

Richard Kraut may not directly write for a positive psychology audience, so one may wonder whether the above discussion is relevant to the topic of the present paper. There are at least two reasons why it is: One is Kraut's prominence as a scholar of Aristotle and in particular of Aristotle's ethics, together with the possible indirect impact of [58]. Concerning the latter, note for example that Daniel Haybron, a philosopher who does speak directly to positive psychologists (e.g., [57]) wrote in his review of Kraut's book [60], that through reflecting on it his views of well-being evolved significantly, with the review containing no hint that he found anything problematic in Kraut's discussion of the oyster.[19] Maybe even more importantly, Kraut's oyster discussion is symptomatic of a more general problem: His discussion fits into a widespread tendency in philosophy-as-usual to make "modally immodest" claims of metaphysical (im)possibilities (in this case, the impossibility of "oyster-like" pleasures *ever* to exceed "human" pleasures) based on what is called the method of cases. This method has been heavily criticised for almost two decades by more empirically oriented philosophers and scientists (summarized for example in "Philosophy within its proper bounds" by Machery [61], see also the critical discussions in *Analysis 80* and *Philosophical Studies 179*, with his replies [62,63]). Unsurprisingly, many philosophers tried to argue against these criticisms. I find these counterarguments unconvincing, but for present purposes this need not be decided here. What is important is that in an interdisciplinary context philosophers present a distorted image of their results when they neglect to at least clearly mention these severe intra-disciplinary doubts about their methodology.

**4. Discussion**

Up to here it has been argued that it is far from obvious that philosophy as usually practiced in this context is helpful to the psychological study of wellbeing, by providing examples where philosophers failed to perform what is widely claimed to be among their core competencies (clarifying concepts), and by pointing to important doubts within philosophy concerning some of its methods. Psychologists should thus not be too surprised that one philosopher painted a rather bleak picture of what one might call philosophy-as-usual:

> Philosophy could be characterized with only a bit of irony as what is left if you begin with the sum total of human thought and subtract those areas in which clear progress has been made. Matters are even less encouraging when it comes to philosophical ethics. The history of ethics looks like a story of progress only if its main texts are read in reverse chronological order. [64] (p. 1)

The present paper does not intend to prove anything as sweeping as this (in fact, I believe this statement is somewhat exaggerated). Nor does it try to develop a complete philosophical analysis (whether a focused one, such as for example an detailed analysis of flow, or a wide-ranging one, such as of the relationship of philosophy and science). The paper is also not concerned with a discussion or an evaluation of the work of particular *philosophers*, only with *some assertions directed at positive psychologists*. For example, take the statement that the "philosophical core behind [eudaimonic psychology], the ideal in question, [is] well-being as the realization of one's potential. *Flourishing*, in the fullest sense of the term. That, at any rate, is what I will propose in this chapter" [57] (p. 27). This is consistent with Haybron's examples of a Comanche warrior (p. 30), and a wolf (p. 31), once these are stripped of the romanticising undertone. It may be that in other work Haybron sheds a different light on this issue and makes my critique above irrelevant (as has been suggested by a comment on this manuscript), but it is not the task of a scientist to read the collected works of every philosopher that directs writings at psychologists, just as few would think of reading the collected works of Fisher before calculating a *p*-value[20].

After these disclaimers the reader may wonder what the purpose is. The main purpose is to point out that a certain form of philosophy, which I refer to as philosophy-as-usual (since it seems to be usual in the philosophical contributions that I read in the positive psychology literature) may not be as easily compatible with positive psychological *science* as seems to be taken for granted. The message to psychologists is *caveat emptor* when buying into philosophers' work. The message to philosophers is: Before "philosophy poses questions to psychology" [11], it better pose some serious questions to itself.

That much said, in the spirit of positive psychology, this paper is intended as a *positive* contribution; therefore, a very rough sketch of an idea how philosophy *could* play a *constructive* role in well-being research is presented as an appendix.

## 5. Could the Match Turn Happy?

Rigour, as a means of weeding out errors, is arguably the characteristic feature of science, but science also crucially depends on creativity, on new ideas and new ways of looking at familiar phenomena. Thus, if contributing rigour may not be what philosophy does, then it appears natural to consider whether instead of narrowing one's view, philosophy might help *widen* it in a relevant way. This appendix intends to very briefly outline one specific way that philosophy could contribute by challenging a number of established practices in positive psychology, from the over- and misuse of Aristotle to uncritical reliance on the "convenient and seductive myth" [67] which is the dual-process model in cognitive science.

### 5.1. Absolving Aristotle

So-called virtue ethics had the chance to widen our "Western" horizon, as the paper that is widely seen as its starting point [68] explicitly declared

> that it is not profitable for us at present to do moral philosophy [...] the concepts of obligation, and duty—*moral* obligation and *moral* duty, that is to say—and of what is *morally* right and wrong, and of the *moral* sense of "ought", ought to be jettisoned. (p. 1)

Yet, "as it has climbed to a position of prominence within Anglo-American philosophy departments, virtue ethics has retreated to an increasingly conventional conception of its central message" [64] (p. 3).

The fault is probably not with Aristotle, but rather with the widespread urge to squeeze Aristotle into the "myopic ways in which contemporary scholars, particularly those influenced by the global West, tend to understand "morality" as a system of obligations" [69] (p. 3). As Anscombe [68] (p. 2) pointed out,

> If someone professes to be expounding Aristotle and talks in a modern fashion about "moral" such-and-such he must be very imperceptive if he does not constantly feel like someone whose jaws have somehow got out of alignment: the teeth don't come together in a proper bite.

(Note: A reviewer reminded me that there is interesting work in virtue ethics inspired by Confucian thought. I admit to being guilty of having neglected this aspect, and take that as another piece of evidence for the importance of stepping outside the usual trains of (philosophical) thought.)

### 5.2. The Value of Returning to the Source(s)

Aristotle receives a disproportionate amount of attention in the wellbeing literature. This is surprising from the psychological science viewpoint, as other notions of a well-lived life have proven themselves valuable in psychological practice. For example, stoic (and other ancient) philosophy was an important basis for the development of cognitive psychotherapy [70], the now well-established benefits of mindfulness point to the possible value of studying its Buddhist background, and parallels between the flow concept and Daoism of the classical Chinese philosopher *ZhuangZi* (莊子, also transliterated *Chuang-tzu*)

have been occasionally noted, see [21] (p. 150f), [71] (chapter 2) or [72]. Nevertheless, these philosophies have been badly neglected. Concerning Daoism, a quick search in PsycInfo found 223 entries for positive psychology and Aristotle, against 1 [sic!] for positive psychology and ZhuangZi.[21] The final part of this paper proposes that the philosophy of ZhuangZi not only fits well into positive psychology, but that it can even illuminate some potentially fruitful ways forward.

### 5.3. A Dao (道 "Way") Forward for Positive Psychology
#### 5.3.1. ZhuangZi: The Sage in Flow

Zhang [72] focused on discussing possible relationships and differences between flow and "forgetfulness" in the ZhuangZi. Here, forgetfulness is involved in the meditative practice of "sitting in forgetfulness" (*zuowang* 坐忘), which means things fit together harmoniously with the result that self-egoistic judgment is suspended and the distinction between "this" and "that" is forgotten ([72,76] (p. 20)). Nevertheless, this suspension of self-egoistic judgment does not require sitting still, and is a state of mind that can also be maintained while acting. Indeed, by dropping pre-conceived judgements as to what must be, it allows mindfulness of the subtle aspects of each situation that presents itself, and thereby facilitates appropriate, spontaneous, effortless action. In other words, the sage following ZhuangZi's way of life "constantly goes by the spontaneous and does not add anything to the process of life" [76] (p. 81). This is clearly reminiscent of the colloquial expression "going with the flow", but one has to be careful at this point: it is *not* meant in the sense of mindlessly drifting along. The Daoist's spontaneous behaviour is not heedless, rather it involves close attention to the situation. [76] (p. 12). In other words, it is "not about resigning oneself to life"; rather, it is about "moving *in rhythm with* the Dao 道" [77]. The point is to not pointlessly fight the course of life—akin to the dictum by the famous psychotherapist C.G. Jung that freedom of the will consists of gladly doing what has to be done (Machek [78] speaks of the Daoist "freedom as doing necessary things".).

Discussing possible parallels to psychological flow theory, Zhang [71] presented Mihály Csíkszentmihályi as using the metaphorical expression *flow* to denote an experience that is characterized by feelings of a dynamic fusion of on-going activity that is effortless, fluid, and creative, an optimal experience leading to life satisfaction and joy when one is in such a flow state. While finding much that is congenial between ZhuangZi's philosophy and the Csíkszentmihályi's flow concept, Zhang nevertheless believed that

> the idea of flow in positive psychology focuses on two key factors, namely, skills and goals. Zhuangzi's flow, however, accentuates more the idea of forgetfulness, or to be more exactly, the forgetfulness of both self and the goal.

Furthermore, motivational psychology is claimed by Zhang to recognize a correlation between flow and happiness and to emphasize goal-oriented and choice-making action—hence investing in consciously chosen goals and planning according to desired results—instead of operating on "random" actions. This supposedly contrasts with ZhuangZi telling us that flourishing consists of more than feeling happy, and that it goes beyond the nature of goal choice or optimizing the effectiveness of action.

The discussion at the beginning of the present paper implies that flow theory does not put as much emphasis on goals as is often assumed, hence ZhuangZi's spontaneous action is even more akin to (an appropriate reading of) Csíkszentmihályi's flow concept than would appear from [72]. Again, using flow in the sense of autotelic harmonious activity, this seems a perfect fit for the Zhangian *yóu* 遊 [79] (p. 545).

#### 5.3.2. Harmony and Flexibility

More subtle are the relationships to positive psychology more generally, and to motivational psychology. In both areas of psychology there are indeed theories that put heavy emphasis on goal choice and achievement, for example goal setting theory [80], the hope theory of [81], or grit [82]. Yet, a number of theories de-emphasize the importance of

cognitively determined goals, for example in cases when they are incongruent with implicit (i.e., subconscious) motives [83,84], or when they are pursued in an obsessive and inflexible way [85,86]. In fact, in one modern and empirically well-supported psychotherapy method, the ability to respond flexibly to circumstances ("the ability to contact the present moment more fully as a conscious human being, and to change or persist in behavior when doing so serves valued ends") is considered the hallmark of mental health [87]. This need not mean that ZhuangZi's ideal sage will never use intellect or will never set goals. Rather, it is about acting without inner conflict and without rigidity; i.e., the "resilient, intelligent, flexible, and creative exercise of agency in response to changing circumstances" [79] (p. 562).

Viewed this way, ZhuangZi's take on a *well-lived* life seems compatible with today's psychological knowledge. The remainder of this paper will argue that paying attention to ZhuangZi's philosophy can also play a constructive role in positive psychology by pointing to ways forward.

### 5.3.3. Way-Making

Today's epidemic levels of excessive stress[22], also in the natural environment[23], seem to indicate that instead of a simple reordering of preferences, a *dysbalance* between the active and contemplative aspects of life—a vita *hyper*activa—may have developed. This extends to flow research as many flow researchers insist on flow being necessarily tied to activities. The present considerations may serve as a corrective in this respect: there is no reason why inner harmony and an autotelic *experience* need involve an *activity* in the ordinary sense.[24]

Even in cases where activity takes place, the ideal is *wu-wei* 無為: effortless action, that is a state of personal harmony in which actions flow freely and instantly from one's spontaneous inclinations—without the need for extended deliberation or inner struggle— and yet nonetheless accord perfectly with the dictates of the situation at hand, display[ing] an almost supernatural efficacy [ . . . ] [91] (p. 7)

This is a well-known feature of ZhuangZi's philosophy. Slingerland argued that attaining some form of *wu-wei* is also a central ideal of philosophers including Confucius, Laozi, Mencius, and Xunzi. [91] (p. 5) Nevertheless, it "plays a more dominant role in the thought of Zhuangzi than in any other major pre-Qin thinker" [91] (p. 175). In [71] (chapter 2) he compared *wu-wei* to flow, but committed the widespread error of confusing enabling conditions ("precise calibration of challenge and skill") with the flow experience itself.

Another feature of Slingerland's account that is worth discussing is his use of the popular dual-processing models of cognition (e.g., [92,93]) in his discussion of *wu-wei*. Similarly, Järvilehto [32] discussed the flow experience in the context of these models, arguing that "while intuition concerns thinking and generating ideas, flow concerns action. Therefore, it can be roughly argued that intuition is System 1 thinking, whereas flow is System 1 doing". Nevertheless, he also noticed that to properly capture the phenomenology of flow, one has to distinguish between different aspects of system 1 processing (which he refers to as instinct and intuition). Similarly, researchers of creativity found it helpful to discuss flow in the context of dual processing, but found that the match is not perfect. [94,95] To finish this paper, I want to very tentatively suggest two points about this:

On the one hand, concerning deliberation and beliefs, relevant for human flourishing may not be so much the *what* than the *how*. It was already mentioned above that psychological flexibility is considered crucial in Acceptance and Commitment Therapy[25] in Rational Emotive Behavioural Therapy also, the dogmatic inflexibility of a patient's demands and musts are what are considered problematic, more so than their contents. In as far as deliberations are about possible courses of action, this fits nicely with the observation by Zhang that "Zhuangzi's recommendation of good life focuses on "how to do" (i.e., following the patterns, processes, and rhythms of the *dao*) rather than "what to do" (i.e., what things that worth doing [sic])" [72] (p. 82) and with the observation by [23] (p. 826) that people are happy not because of what they do, but because of how they do it. As a well-known passage in the ZhuangZi explains, this means not allowing likes and dislikes to damage

you internally, instead making it your constant practice to follow along with the way each thing is of itself [*ziran* 自然], going by its spontaneous affirmations, without trying to add anything to the process of generation [97] (p. 51).

ZhuangZi contrasts this to the behaviour of those who "treat [their] spirit as a stranger", waste their energy, and use their nature-given body to quibble about abstract questions such as whether "hardness" and "whiteness" can exist in separation from one another. (ibid) This can serve as a warning against over-intellectualization in positive psychology. Additionally, facing up to how things are "in themselves" and following along with what is happening presupposes an absence of concern about being in control. This shows how important it would be for positive psychology to properly take into account that the flow experience involves an *absence of worry about losing control*, rather than the presence of a sense of control as usually maintained (compare [21] (p. 59)).

Finally, continuing the point above concerning dual process models: they are undeniably popular, but it seems too much to say that there is " general agreement that human thought is characterized by two distinct systems" [71] (chapter 1). Not only have Dijksterhuis et al. raised doubts, as acknowledged by Slingerland [71] (endnote 15 to chapter 1), but, for example, Stanovich [98] argued for a tripartite model of cognition in which type 2 processing is subdivided into an algorithmic and a meta-cognitive form of processing, and there are views in psychotherapy such as the "smart unconscious" of Milton Erickson, or the "felt experience" in the psychotherapeutic technique called focusing, which was developed by the philosopher Eugen Gendlin [99]—effectively arguing for a partition of type 1 processes into at least two subtypes.[26] This naturally leads to the idea that one could subdivide both type 1 and type 2 processing, and indeed there is at least one psychological theory which does that: personality systems interaction theory [102–105]. In this theory, optimal human experience and a flourishing life are posited to depend on the proper *cooperation* of four cognitive systems, *including both a conscious–deliberative and an intuitive–holistic high-level* system. This in turn seems to realize what Slingerland [71] views as "[t]he goal of *wu-wei*[:] to get these two selves [conscious and unconscious cognition] working together smoothly and effectively". Indeed, putting more emphasis on refining dual-process models and on the importance of integrating the various processes might be another way forward for positive psychology.

**Funding:** The APC was funded by the University of Vienna.

**Data Availability Statement:** Not applicable.

**Conflicts of Interest:** The author declares no conflict of interest.

## Notes

1.  The claim that humanistic psychology was deficient in terms of rigorous science was disputed and later partly retracted, see [2] (p. 18). However that may be, this dispute is irrelevant for the present work as the latter is concerned with the relationship between philosophy and the scientific study of wellbeing in positive psychology.

2.  I use this term for lack of a better one. What I find problematic in philosophy does not neatly line up with the popular analytic versus European distinction. The phrase is chosen in parallel with "business as usual", which is a way of saying that things continue as normal. Of course, this can be a positive thing (e.g., being able to keep things going without interruption during times of crisis) or negative (e.g., insufficient flexibility to adopt to changing circumstances). I leave it to the reader to determine in which respects and to what extend philosophy-as-usual is to be seen positively, negatively, or neutral.

3.  It is provided as an appendix since I am painfully aware of the brevity of my discussion. Nevertheless, I felt a work related to *positive* psychology should at least sketch a positive outlook; a fuller treatment has to await another occasion.

4.  And possibly also for the psychology of judgement and decision making, see the last section of this paper.

5.  Adkins [15] (p. 297): "*aretê* denoted and commends "excellence", not "virtue""; Urmson [16] (p. 30): "*aretê*: excellence or goodness of any kind. [...] *aretê* is commonly translated virtue, a transliteration of the Latin *virtus*, but neither *aretê* nor *virtus* means virtue, except in such archaising expressions as "the virtues of the internal combustion engine".

6.  Aristotle seems to see this the same way, see EN 1177b31-1178a4.

7.  These appeared after the first draft of the present manuscript was written.

8     All italics in quotes are in the original, unless otherwise indicated.

9     An invention that seems to have come only after the development of complex societies [42]

10    I.e., appealing to a pessimistic induction on ethical theories analogous to the pessimistic induction in the philosophy of science.

11    For example, in the famous paper by Hardie [54].

12    The second motivation they provide is the claim that the perception of science as value-free is misconceived—a claim that seems itself misconceived, see the previous subsection.

13    This problem seems somewhat similar to that often assigned to life-satisfaction accounts of wellbeing: that lowering expectations makes satisfaction easier to achieve. In fact, here it appears to be more serious, since adaptive expectations often make sense (overly unrealistic expectations tend to be bad for you), whereas lowering one's potential seems dysfunctional in general.

14    I for my part can not even imagine how locusts taste to humans, even though they are being eaten by many.

15    About 1,504,000,000,000,000,000,000,000 according to https://www.quora.com/How-much-sand-is-in-Sahara-desert, accessed on 12 May 2022.

16    Daniel Dennett's abbreviation for "Very much more than ASTronomical", used to refer to "finite but almost unimaginably larger numbers than such merely astronomical quantities as the number of microseconds since the Big Bang times the number of electrons in the visible universe" [59] (ch.6 endnote 36).

17    Again this undefined "we."

18    Note that Kraut does not restrict to realistic life spans. E.g., "There is nothing wrong, then, with McTaggart's idea of a human life that lasts a million years" [58] (p. 228).

19    Personally, I find this particularly unfortunate since I do sympathize with much of what Kraut and also Haybron write. There *might* be something relevant to science in there; but as a scientist, I dislike the prospect of having to double check which of philosophers' alleged arguments are actually reliable.

20    The use of *p*-values in psychology is controversial, but this does not invalidate the point: the difference is that statisticians have pointed to the intricacies in using p-values for decades but were widely ignored by psychologists (e.g, [65,66]), while philosophy-as-usual sweeps its problems under the rug.

21    Admittedly, there have been a number of studies of mindfulness in positive psychology; but see [73,74] for lacunas in this area and for the under-explored complications in the relationship between Buddhism and science. Additionally, parallels between the flow concept and the philosophy of the ZhuangZi have been noted occasionally (e.g., [21] (p. 150f), [75,76]), but not discussed in the context of the relationship between philosophy-as-usual and positive psychology.

22    A possible translation of *dao* [88] (p. 57).

23    Which apparently was among the original reasons to develop flow theory, compare the introduction to [20].

24    Examples of optimal experience without action may be provided by meditative absorptions (*jhānas*), as even the first stages involve intense joy (*pīti*) and happiness (*sukha*) without anything that one would ordinarily call an activity. Compare, e.g., [89,90]. Aristotle would seem to agree: EN 1154b 26-28.

25    See [96] for a brief introduction and philosophical consequences.

26    Other prominent psychological work critical of a simplistic dual-process model includes [67,100,101].

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
