# Peer review of "Positive Psychology and Philosophy-as-Usual: An Unhappy Match?"

_philosophies, doi:10.3390/philosophies7030052_

Round 1

Reviewer 1 Report

This is a creative, engaging and thought-provoking paper.  While there are areas that can be clarified, tightened, and developed, the paper's critique of other authors and analysis of the limitations of certain applications of philosophy to positive psychology make it a worthwhile contribution.  There are a few issues with the writing that need to be addressed (corrections have been sent to the editors), but the author's writing is generally clear and well-organized.  Here are a few suggestions - 

"Philosophy as usual" is a phrase the author uses frequently, but it is not clear what it means.  Given the diversity of contemporary philosophy and the author's emphasis on precise definition, it would be helpful to define that term here.  

The author does a nice job showing problems with the way that "flow" is conceptualized, and they make helpful distinctions in the treatment of the concept (e.g., between preconditions of flow and the nature of the experience itself).  

In the paragraphs at the end of page 8, the author seems to move from a critique of conceptions of normativity (or failures to precisely define normativity) to a critique of the substance of the moral arguments of Aristotle, Kant, etc.  One can almost always find specific elements of a thinker’s moral philosophy that are problematic, but that’s a different problem than the one that the author is speaking about earlier (i.e., failures to distinguish different kinds of normativity).  The author should better clarify how the failure of particular moral theories, or the issue of relativism in ethics (which is an aspect of what he’s discussing here), relates to the definitional problem he is discussing.  The shortcomings of particular ethical theories is different from the failure of conceptual clarification and rigor.  The relationship between the two needs to be addressed.

The section on Kraut’s oyster is thought-provoking.  The author might distinguish between their specific objections to the oyster example (e.g., the inability to imagine the consciousness of an oyster) with the usefulness of thought experiments at all (in particular, their relevance to positive psychology).  Zhuangzi, for instance, has a passage about positively evaluating the experience of a fish.  Is this equally vulnerable to criticism, or does the way that Zhuangzi engages in this thought experience (in his usual subversive, playful way rather than with the presumed rigor of an analytic philosopher) make it more useful? 

It is interesting that the thinker who the author puts forward as worthy of our attention, Zhuangzi, is someone who rejects conceptual rigor completely.  His views of language are relevant here.  The author might address how his focus on the need for definitions and conceptual rigor throughout the paper relate to his praise for a thinker who revels in subverting the entire project of precisely defining terms.

Given the large amount of excellent work on virtue ethics (not only work based on Greek thought but also Confucian thought) the paragraphs on the top of p. 13 far too quickly dismiss this approach to ethics.  Many would disagree that philosophers “squandered” the opportunity to widen our horizon (on the bottom of page 12, the author might rethink the phrasing of the sentence that implies that “virtue ethics…squandered” its chance.  Philosophers can squander a chance, but an entire approach to ethics cannot do any such thing.)

I highly recommend that the author look at Edward Slingerland’s book “Effortless Action,” which is all about the concept of Wu Wei, a key concept in Zhuangzi connected with the flow state.

For Zhuangzi, effortless action (“flow”) is connected with a notion of following one’s nature rather than conscious deliberative thought (tian vs. ren).  The author should bring in some concept of "according with nature" here, as it’s essential to understanding Zhuangzi.  Zhuangzi's understanding of nature explains why his concept of flow, while requiring the setting aside of consciously goal-directed behavior, is not random.  It is based on ziran, what is “so of itself,” or what something receives from nature (qi suo shou). 

Reviewer 2 Report

 Review of Positive Psychology and Philosophy-as-usual: An Unhappy Match?

I fundamentally like this paper and think it could be an important addition to the Philosophies Special Issue on Wellbeing. Among the most valuable contributions it makes is an attempt to disambiguate of the concept of flow. But it covers a very broad field of topics and theories and make a very large number of claims and observations, most of which are interesting, but some of which are also quite controversial and perhaps overly strong. I would generally recommend the author to reduce both the number of specific targets for his/her criticism (alleged instances of “philosophy as usual” and the amount and degree of polemics). The trouble with philosophy as usual seems to be a very diverse set of faults, from not being sufficient attentive to the original context of a philosopher’s work, over making overly strong claims as to the mutual exclusiveness of flow and virtues activities and employing a problematic distinction between subjective and objective wellbeing, to not making explicit what is meant by “normative”.

The introduction of ZhuangZhi’s thinking towards the end of the paper is interesting and adds a cross-cultural perspective on the topic which would complement the other contributions to the special issue well. However, it comes in a late stage of the investigation; moreover, it might seem as though the author applies different standards of “conceptual rigor” in different parts of his investigation, criticizing contemporary philosophers who are hardly less rigorous or precise in their use of concepts than this ancient Chinese philosopher. The distinction between science and (seen as more problematic) philosophy also seem exaggerated.

  1. 2 line 78ff. Better write terms (like “happiness”) in “ “, to distinguish them from concepts

  1. 1-3 The introduction is very conversational in style, and contains (too) many meta-comments, for example about whether or not the paper is an intercultural comparison etc. The paper would benefit from a more concise style of writing and presentation

  1. 7 section 2.2. The discussion of objectivism about wellbeing is interesting and to some extent insightful. But the criticism of the standard way of distinguishing subjective and objective WB seems overblown. MacLeod's way of distinguishing does seem very intuitive. The case of subjective appraisal based on post-traumatic growth IS covered by (the second disjunct in) MacLeod's definition – it is about a subjective attitude, a positive evaluation of a state of affairs, and need not, on MacLeod’s view, be based on “positive feeling states”. The author rightly points to the broad scope of the notion of objective wellbeing, and ways in which “external standards” could be considered subjective. But this does not suffice to debunk the standard philosophical distinction between s. and o. WB (it does make sense to define as “objective” everything that is not part of a subject’s mental states or determined by her subjective attitudes, even if it may be “subjective” in another sense.

  1. 8: The Wittgenstein example is a counterexample to the concordance thesis, because W.s (positive) subjective judgment did not correspond to his (negative) objective state (of non-flourishing)

  1. 8. Section 2.3. The author touches on some very big questions here. There’s a huge literature on how normative force or the moral or prudential “ought” should be understood. It is not that philosophy has nothing to say about this, only because it is not explicitly thematized in work on well-being.

Line 456: “Baldy” > “badly”

  1. 10 above: The criticism of Haybron’s way of understanding eudaimonia is overblown. The idea is that there is a common human potential (probably not quantifiable); a set of ideal human qualities and activities. Genghis Khan did not realize these, as he for example may have lacked temperance, modesty, wisdom etc.)

  1. 11. The criticism of Kraut's Oyster example is not uninteresting or completely unfounded, but it sounds exaggerated (and, as other parts of the paper, too polemical).

  1. 14 How relevant are the qualities of Zhuang Zi's "ideal sage" to the well-being of ordinary, not particularly ideal people?

Line 700ff. It may be too much to also bring Hanna Arendt into the discussion. That said, the point that flow and autotelic experiences need not involve (here mistakenly written as “involvement”) activities in the ordinary sense is a very good and important one.

Line 709: I am generally sympathetic to the critical discussion of dual process psychology. But it complicates the investigation even more. Besides, it does sound as if the author at some points commits him/herself too strongly to the dual process model. The author is right Aristotle stresses deliberation; but his view of deliberation is not one-sidedly contemplative, and it is debatable both whether the System 1 and 2-distinction applies to this case and whether it is fitting, quite generally (as there seems to be many processes that fit neither the one nor the other system completely). The author eventually agrees with the latter view, but the “first glance”-observation makes it difficult for the reader to grasp precisely what he/she intends and claims.

Author Response

This manuscript is a resubmission of an earlier submission. The following is a list of the peer review reports and author responses from that submission.